# OFFLINE DEEP MODELS CALIBRATION WITH BAYESIAN NEURAL NETWORKS

## ABSTRACT

In this work the authors show that Bayesian Neural Networks (BNNs) can be efficiently applied to calibrate state-of-the-art Deep Neural Networks (DNN). Our approach acts offline, i.e., it is decoupled from the training of the DNN to be calibrated. This offline approach allow us to apply our BNN calibration to any model regardless of the limitations that the model may present during training. Note that this offline setting is also appropriate in order to deal with privacy concerns of DNN training data or implementation, among others. We show that our approach clearly outperforms other simple maximum likelihood based solutions that have recently shown very good performance, as temperature scaling (Guo et al., 2017). As an example, we reduce the Expected Calibration Error (ECE%) from 0.52 to 0.24 on CIFAR-10 and from 4.28 to 2.46 on CIFAR-100 on two Wide ResNet with 96.13% and 80.39% accuracy respectively, which are among the best results published for these tasks. Moreover, we show that our approach improves the performance of online methods directly applied to the DNN, e.g. Gaussian processes or Bayesian Convolutional Neural Networks. Finally, this decoupled approach allows us to apply any further improvement to the BNN without considering the computational restrictions imposed by the deep model. In this sense, this offline setting is a practical application where BNNs can be considered, which is one of the main criticisms to these techniques. In terms of reproducibility, we provide all the implementation details in https://github.com/2019submission/bnn.2019.

## 1 INTRODUCTION

Deep Neural Networks (DNNs) have achieved state of art performance in many task such as Image Recognition (Huang et al., 2017; Szegedy et al., 2017; Zagoruyko & Komodakis, 2016), language modeling (Mikolov et al., 2013a;b), machine translation (Vaswani et al., 2017) or speech (Hinton et al., 2012). For that reason, neural networks are now used in many applications. However, this state-of-the-art performance is measured in terms of accuracy, but there are many tasks in which the probabilistic information must be also *reliable*. For example, a probabilistic classifier can be incorporated into a more complex model considering multiple sources of information, by the use of e.g., probabilistic graphical models (Koller & Friedman, 2009), or by combining neural networks with language models in natural language processing tasks (Gulcehre et al., 2017). In addition, probabilistic outputs of classifiers have proven to be useful in many areas apart from classical machine learning tasks, such as language recognition (Brümmer & van Leeuwen, 2006), language models for speech recognition (Tüske et al., 2018) or medical diagnosis (Caruana et al., 2015).

In Bayesian statistics, the reliability of probabilities is measured by their calibration. As a consequence, the machine learning community has been exploring methods to calibrate the output of classifiers to achieve the many beneficial properties of well-calibrated probabilities (Zadrozny & Elkan, 2002a; Cohen & Goldszmidt, 2004; Niculescu-Mizil & Caruana, 2005). Nowadays, there is an increasing interest in obtaining reliable probabilities in the deep learning community. In the past, neural networks trained with a cross-entropy criterion tended to present relatively good calibration. However, a relevant recent work in Guo et al. (2017) has evidenced that modern state-of-the-art neural networks are badly calibrated in general. Moreover, the same work shows that calibration can be dramatically improved by very simple maximum-likelihood parametric techniques, among which Temperature Scaling (TS) is highlighted as the preferred choice, due to its extreme simplicity,

**ResNet-101: 93.46% accuracy (CIFAR10)**

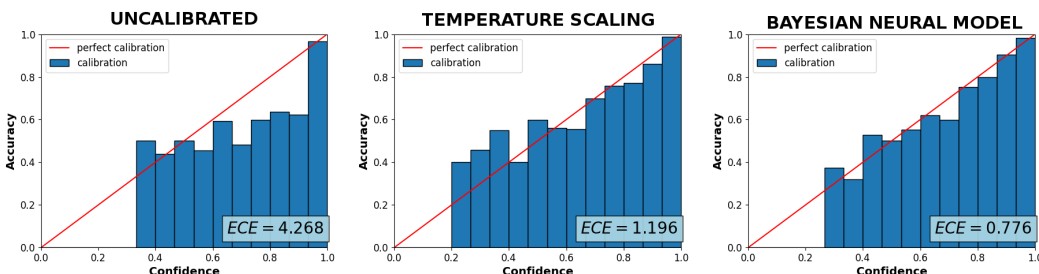

**WideResNet-40x10: 76.74% accuracy (CIFAR100)**

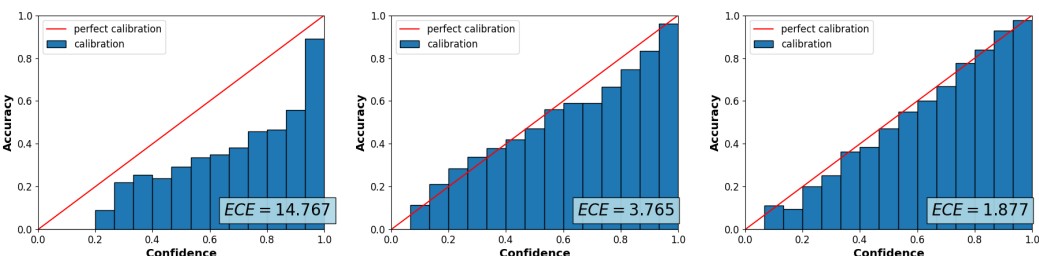

Figure 1: Reliability diagrams (Guo et al., 2017) for two networks trained on CIFAR-10 and CIFAR-100. The red line represents perfect calibration. We plot the Expected Calibration Error (ECE %) for 15 bins (see section 5 for a description). The lower the better.

very good behavior in general and computational efficiency. In fact, TS outperforms more complex techniques in most cases, leading to the conclusion that good calibration can be better achieved with simpler techniques. This conclusion follows the hypothesis that the space configured by the outputs of a deep model is relatively simple, and therefore good performance, measured as Expected Calibration Error (ECE), can be achieved with very simple models. In fact, TS is a technique that performs nicely in complex multiclass tasks (Guo et al., 2017).

In general, there are two main approaches to reduce overconfidence: implicit or online and explicit or offline. An implicit method aims at obtaining calibrated distributions directly at the output of the model, while explicit methods post-process the output of the model to be calibrated. In this paper we propose an offline method based on a Bayesian Neural Networks (BNNs) to obtain calibrated probabilities, see figure 2. We use BNNs as we aim at being benefited from two key properties of Bayesian statistics and neural networks: the high expressiveness of neural network models and the capabilities of the Bayesian statistics to model the uncertainty. We assume the hypothesis that as long as the uncertainty is correctly modelled, we can use high expressive function approximators for the task of calibration. These high expressive models are required since we assume that the calibration space is not simple. However, the generalization capability of these models are achieved through proper uncertainty consideration. Figure 1 shows reliability diagrams comparing our BNN method to TS.

This work is organized as follows. We first provide an insight on why Bayesian Statistics and neural networks are suitable to adjust confidence in output probabilities. We then describe our offline calibration approach based on BNNs. We end up comparing our method to TS and reporting clear performance improvements. Finally, we discuss our approach against recent published techniques, enumerate some beneficial properties and propose possible improvements over this contribution.

## 2 BAYESIAN MODELLING AND CALIBRATION

In a classification scenario, calibration can be interpreted as the agreement between the probabilities of a class assigned by a model to a set of samples, and the proportion of those samples where that class is actually the true one.

One way of achieving calibration is to reliably modelling the probabilistic distributions of the data from the classes involved. This is the main strength of Bayesian models, which manage uncertainty properly, in contrast to point-estimate approaches (i.e. Maximum Likelihood or Maximum Posterior). In the former, the uncertainty is incorporated by taking an average of all the likelihood models under the posterior distribution on the parameters, given an observed set of data:

$$p(t|x) = \mathbb{E}_{p(\theta|\mathcal{O})}\{p(t|x,\theta)\},\tag{1}$$

where $\theta$ are model parameters, $x$ represent a sample for which we want to predict a label $t$ and $\mathcal{O} = \{(x_i, t_i)\}_{i=1}^N$ is the set of observed samples[1]. In Bayesian approaches, it is indeed the observed data what model how representative a likelihood model is for a particular task, and thus how it influences the predictions. On the other hand, in point-estimate models all the decision is based on a choice of the parameter once the model is trained.

For instance, consider the case of a MAP network (e.g., a typical deep convolutional model trained with cross-entropy loss and $L_2$ regularization). This model explains the data based on a point-estimate training, i.e., by representing what is more likely to appear. Nowadays, this gives outstanding accuracy in classification tasks, but it is easy to train an over/under-confident model, i.e., the one that outputs too extreme probabilities, even for unfavourable cases like e.g. when data that has conditions not unseen in the training set, or not explained by the expressiveness of the model itself. This could happen if the true distribution does not lie in the family of parametric models $p(t|x,\theta)$. We will refer to both conditions as *mismatch*. This is very harmful for the calibration, because in those mismatch cases, the model should yield more moderate probabilities, otherwise the classification errors will be more catastrophic. In other words, in tasks where calibration matters, a classification error has unequal consequences if the probabilities are moderate or extreme. Thus, for example, if there exists such mismatching conditions, what is likely to happen, a point estimate will not represent the data (i.e. the probability assigned) in the way it should, possibly leading to over-or under-estimation of probabilities. On the other hand, in the Bayesian framework, a posterior distribution on the parameters could consider networks explaining these mismatching conditions. By averaging different contributions, the model ideally moderates probabilities on unfavourable data. For the sake of illustration, we provide a simple example in appendix A. We encourage unfamiliar readers with Bayesian statistics and calibration to read this appendix.

Regarding the accuracy of Bayesian models, in Bayesian decision theory, if the model used to generate the data is known, the optimal error can be achieved, which also means optimal accuracy if all decision costs are equal. This suggests that a proper way of assigning probabilities is also paramount for the accuracy. Thus, by choosing appropriate densities for the class-conditional probabilities $p(t|x)$, such as factorized multivariate Bernoulli distributions like in point estimate models, the accuracy will also be correctly modelled. Moreover, it is well known that Bayesian models asymptotically tend to point estimate models as the data increases in size, see Duda et al. (2000) section 3.4. Therefore, it is expected that good accuracy performance achieved by point-estimate models should be also achievable by Bayesian models, at least for sufficient amounts of data.

## 3 OFFLINE CALIBRATION WITH BAYESIAN NEURAL NETWORKS

The architecture proposed in this work is shown in Figure 2. We apply a BNN to the potentially uncalibrated outputs of a DNN model. The goal is to improve the calibration minimizing the accuracy degradation of the original DNN model. Our approach works offline, meaning that given a DNN model, we project the available data to the space defined by the outputs of the model in the form of logit, i.e. pre-softmax values. This new representation is then the input to our BNN. The BNN aims at taking this uncalibrated so-called *logit space* and project it to a new calibrated one. The same procedure is applied for the TS method.

---

[1]This is a particularization of Bayesian learning for class-conditional modelling.

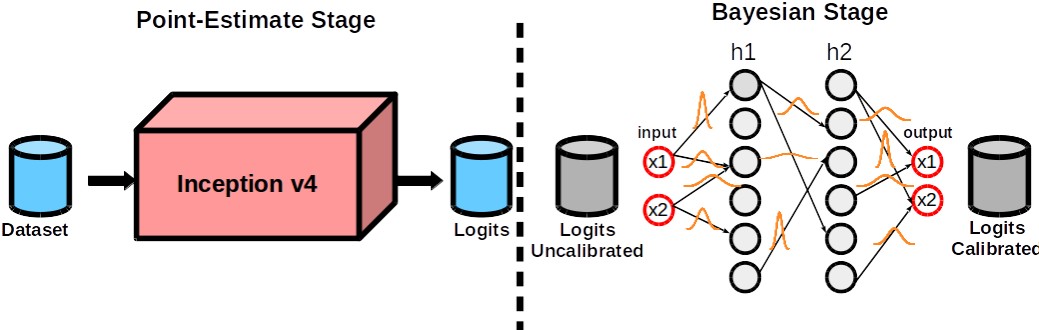

Figure 2: Exemplary representation of the architecture of our proposed model. On the left, an expensive pretrained DNN on ImageNet is trained on a specific dataset (transfer learning). Then, the (uncalibrated) output of such DNN is the input to the BNN calibration stage. This stage is trained by the maximization of the Expected Lower Bound (ELBO) and predictions are done using Monte Carlo integration. The inputs and outputs of the Bayesian stage have same dimension (given by the number of classes), and lie in the so-called logit space. Orange Gaussians on each arrow represent variational distributions on parameters. We do not plot all the arrows for clarity. This Bayesian stage is independent of the previous one as we only require access to the logits of an already trained model.

This off-line set-up presents clear advantages. First, the approach is efficient, since the DNN model does not need to be re-trained for re-calibration. Furthermore, we can incorporate future improvements to the BNN calibration stage without affecting the previous DNN model. Second, our proposal is extremely flexible, as the proposed BNN calibration stage will work with any probabilistic model, even implicitly-calibrated models, with potential benefits on calibration performance. And finally, our proposal preserves privacy, because there is no need to access the original data used to train the DNN model, or even the DNN topology: to be trained, the BNN only needs the data projected to the outputs of the DNN on the logit space, and the original targets $t$.

For these Bayesian approaches one has to compute the posterior distribution $p(\theta|\mathcal{O})$ and the expectation in equation 1. Using configurations that yield to analytic solutions to both problems result in low-expression models for this task. We solve this problem by choosing Neural Networks to parameterize the likelihood $p(t|x,\theta)$ of our BNN, and therefore taking advantage of the high expressiveness of these models. In this case, several intractabilities arise that must be addressed.

In this work we approximate expectations on equation 1 with Monte Carlo integration, and the posterior is approximated by a variational distribution in terms of the Kulback-Lieber Divergence. This is done by the maximization the Evidence Lower Bound (ELBO). We use stochastic optimization based on the reparameterization trick (Kingma & Welling, 2014; Rezende et al., 2014) to approximate the expectation under the variational distribution, and also mini-batch stochastic optimization for expectations under data distribution. Thus, our training criteria is given by:

$$ELBO = \frac{1}{N} \sum_{(x,t)\sim p_d(x,t)} \Big[ \frac{1}{K} \sum_{\theta\sim q_\phi(\theta)} [\log p(t|x,\theta)] - \beta \cdot D_{KL}\{q_\phi(\theta)//p(\theta)\}\Big], \qquad (2)$$

where we introduce $\beta$ following Blundell et al. (2015).

Our variational distribution is a factorized Gaussian distribution and for that reason we refer to our BNN approach as a *basic* approach, as we do not incorporate any improvement recently proposed for BNN models, such as normalizing flows, local reparameterization, and so on. Also, we use a standard normal density for the prior. We choose this simple approximation to demonstrate our starting hypothesis: that BNNs can be applied to improve the calibration of state-of-the-art DNN in a very efficient way. Our basic BNN model can be improved by using normalizing flows (Rezende & Mohamed, 2015; Kingma et al., 2016; Huang et al., 2018; van den Berg et al., 2018), auxiliary variables (Agakov & Barber, 2004; Ranganath et al., 2016; Maaløe et al., 2016), local reparameterization (Kingma et al., 2015), combinations of all of them (Louizos & Welling, 2017) or deterministic models (Wu et al., 2018). Also, Cremer et al. (2018) has recently pointed out that

amortized inference leads to an additional gap in the bound, in addition to the $D_{KL}$ gap between the true and variational posteriors; and we can also use other proposals to mitigate this effect (Shu et al., 2018; Kim et al., 2018).

Finally, class predictions are assigned by first computing the logits of a test sample using the first DNN stage $\mathcal{B}$, and then using them as inputs of our BNN to yield calibrated probabilities, which can be formalized as follows:

$$l = \mathcal{B}(x)$$
$$p(t|x,\mathcal{O}) \approx \frac{1}{M} \sum_{i=1}^{M} p(t|l,\theta_i)\,;\ \theta_i \sim q_\phi(\theta), \tag{3}$$

where $M$ is a value chosen on validation. Note that our proposed BNN is not as efficient as TS for calibration. However, the contributions to the weighted average can be fully parallelized computationally, as predictions do not depend on each other. Thus, we can use modern GPU libraries such as CUBLAS and batch-based operations to dramatically speed-up the process.

## 4 RELATED WORK

To our knowledge, TS (Guo et al., 2017) has been consistently reported as the best technique to improve calibration over a list of classical ways of improving calibration, such as histogram binning (Zadrozny & Elkan, 2001), isotonic regression (Zadrozny & Elkan, 2002b), Platt scaling (Platt, 1999) or Bayesian binning into quantiles (Naeini et al., 2015) among others. For a recent description and performance comparison with modern neural networks, see Guo et al. (2017). On the other hand, there are several works that study overconfident predictions and model uncertainty in different contexts, but without reporting an explicit measurement of calibration performance in deep neural models. For instance, Gal & Ghahramani (2015) connect Bernoulli dropout with BNNs, and Gal & Ghahramani (2016) links Gaussian processes with classical dropout regularized networks, showing how uncertainty estimates can be obtained from this networks. In the latter, the authors state that these Bayesian outputs are not calibrated. In Pereyra et al. (2017), an entropy term is added to the log-likelihood to relax overconfidence. Lakshminarayanan et al. (2017) propose training network ensembles with adversarial noise samples to output confident scores. Chen et al. (2018) propose a model that uses probes of the individual layers of the neural network classifier to create a confidence score for the network output. DeVries & Taylor (2018) train a second output obtained from the penultimate layer of the classifier to be confident by interpolation of the softmax output and the true value, scaled by this score. Lee et al. (2018) proposed a generative approach for detecting out-of-distribution samples but evaluates calibration performance comparing their method with normal cross-entropy minimization, using TS as the calibration technique.

On the side of BNNs, Kingma et al. (2015) formalize Gaussian dropout as a Bayesian approach. In Louizos & Welling (2017), novel BNNs are proposed, mixing inverse autoregressive flows (Kingma et al., 2016), auxiliary variables (Maaløe et al., 2016) and local reparameretization (Kingma et al., 2015). None of these approaches measure calibration explicitly on deep neural models, as we do. For instance, Louizos & Welling (2017) and Lakshminarayanan et al. (2017) evaluate uncertainty by training on one dataset and use it on another, expecting a maximum entropy output distribution. More recently, Zhang et al. (2018) propose an scalable inference algorithm that is also asymptotically accurate as MCMC algorithms.

We compare our proposed BNN approach against TS, as to our knowledge it is the state of the art in calibration tasks involving deep neural models, according to the reviewed previous work. TS widely improves calibration without affecting the accuracy, and can be efficiently applied to any model. Formally, given $\mathcal{O}$, TS maximizes the log-likelihood of the conditional distribution $p(t|x/T)$ w.r.t. the parameter $T$. In this case, $x$ also represents the output logits of the deep convolutional model.

In Section 6, we will comparatively discuss the properties and results of our proposal with other works recently found in the literature for calibration on deep models, some of them based on implicit and explicit models.

## 5 EXPERIMENTS

We demonstrate calibration performance on several computer vision models on several datasets. We have used CIFAR10 and CIFAR100 databases (Krizhevsky et al., a;b); SVHN (Netzer et al., 2011); and a GENDER recognition task (Eidinger et al., 2014). We use a validation set randomly taken from the training set with 5000 samples for CIFAR10 and CIFAR100, 10000 samples for SVHN and 4005 for GENDER. This validation set is used to train the TS parameter and to choose the number of Monte Carlo samples, $M$ in equation 2. We report results for the best model on validation for all the tested configurations. We optimize the ELBO using adam optimization (Kingma & Ba, 2014), since it performed better than stochastic gradient descent on our previous experiments. We used $\beta = 0.1$ from the set $\{1, 0.1, 0.01\}$ as it behaves better on validation.

In order to compare our experiments with uncalibrated and TS calibrated probabilities, we used an unbiased estimator of the Expected Calibration Error (ECE) computed as in Guo et al. (2017), with 15 bins. The ECE measures the expected value of the difference between accuracy and confidence:

$$\mathrm{ECE} = \sum_{i=1}^{15} \frac{|B_i|}{N} |\mathrm{acc}(B_i) - \mathrm{conf}(B_i)| \tag{4}$$

where $N$ is the number of total samples; $B_i$ represents the set of samples whose predictions $t$ confidence lie in bin $i$; $\mathrm{conf}(B_i)$ is the average confidence and $\mathrm{acc}(B_i)$ is the accuracy of that bin. We also report the accuracy of our models. This is because a classifier can be perfectly calibrated, but useless from a classification point of view.

Note that we evaluate our proposed method on several state of art configurations of computer vision neural networks over the mentioned datasets: Wide Residual Networks (Zagoruyko & Komodakis, 2016), Residual Networks (He et al., 2016b), Densely Connected Neural Networks (Huang et al., 2017), Pre-Activation Residual Networks (He et al., 2016a), Dual Path Networks (Chen et al., 2017), VGG (Simonyan & Zisserman, 2014) and ResNext (Xie et al., 2017). The results reported in this work are obtained from some pretrained neural networks.

### 5.1 RESULTS

Table 1 shows the results for the different datasets, original DNNs and calibration approaches. We presents results in %, i.e. multiplying by 100 the result obtained in equation 4. The most important point is that calibration is improved by a wide margin in every model except for two models in CIFAR100 and one model in SVHN. Table 1 shows that in average our proposed calibration method outperforms TS with an insignificant accuracy loss. This means that high expressive models can cope with the calibration task as long as uncertainty is correctly modelled. Therefore, we propose an alternative hypothesis to the one given in Guo et al. (2017) where the authors argued that the calibration space is simple. We argue that if highly complex models outperform simple ones is because the distribution of the calibration space is also complex but the low dimensionality of the logit space makes high expressive models overfit.

We realized that more expressive models are needed by more complex tasks, like CIFAR100. For instance, ResNet-18 GENDER uses BNNs of two layers with two neurons per layer, while WideResNet 40x10 on CIFAR100 uses two layers of 2000 neurons. This reflects that, when dimensionality increases, more expressiveness is needed. Another important point observed in SVHN (see Densenet-169, ResNet-50 and WideResNet 16x8) is that TS has degraded calibration by a factor of three in the worst case. In general BNNs do not degrade the calibration.

One drawback of our basic approach is that in some cases we obtain slight accuracy degradation. Accuracy degradation is more relevant only for CIFAR100, however our BNN method reduces ECE15 by a factor of two in some experiments in this task. Moreover, in some cases we are able to improve both, accuracy and calibration, see WideResNet 40x10 for CIFAR10 or ResNet-18 for GENDER dataset. Thus, we cannot conclude that BNNs are calibrating at the cost of losing accuracy. This motivates us towards further research on this accuracy degradation, as we expect to solve it with more sophisticated approximations in future work. Possible hypothesis for this degradation are that either the gap between the variational and the true posterior is still large, the variance of the ELBO estimator is large and does not allow us to converge to a better optimal, or the expressiveness of the likelihood model is not enough to deal with the particular logit space distribution. We found that for

Table 1: ECE 15(%) and Accuracy (%) comparing model uncalibrated, calibrated with TS and with BNN

| | **CIFAR10** | | | | | |
| | uncalibrated | | Temp Scal | | BNN | |
| | Acc | ECE | Acc | ECE | Acc | ECE |
| --- | --- | --- | --- | --- | --- | --- |
| WideResNet 28x10 | 96.13 | 1.835 | 96.13 | 0.518 | 96.08 | 0.243 |
| DenseNet 121 | 95.49 | 2.643 | 95.49 | 1.011 | 95.26 | 0.600 |
| DenseNet 169 | 95.49 | 2.664 | 95.49 | 0.826 | 95.29 | 0.511 |
| Dual Path Network 92 | 95.18 | 2.995 | 95.18 | 1.072 | 95.03 | 0.730 |
| ResNet 101 | 93.46 | 4.268 | 93.46 | 1.196 | 93.38 | 0.776 |
| VGG 19 | 93.68 | 4.412 | 93.68 | 1.708 | 93.67 | 0.843 |
| Preactivation ResNet 18 | 94.93 | 3.155 | 94.93 | 0.570 | 94.73 | 0.455 |
| Preactivation ResNet 164 | 93.91 | 4.102 | 93.91 | 0.437 | 93.82 | 0.331 |
| ResNext 29_8x16 | 94.79 | 2.833 | 94.79 | 0.741 | 94.61 | 0.728 |
| Wide ResNet 40x10 | 95.01 | 3.001 | 95.01 | 0.921 | 95.08 | 0.594 |
| **average** | 94.81 | 3.191 | 94.81 | 0.9 | 94.70 | 0.581 |

| | **SVHN** | | | | | |
| | uncalibrated | | Temp Scal | | BNN | |
| | Acc | ECE | Acc | ECE | Acc | ECE |
| --- | --- | --- | --- | --- | --- | --- |
| WideResNet 40x10 | 96.95 | 1.26 | 96.95 | 1.17 | 96.90 | 1.15 |
| Densenet-121 | 96.76 | 2.021 | 96.76 | 1.092 | 96.69 | 0.716 |
| Densenet-169 | 96.70 | 0.363 | 96.70 | 1.016 | 96.59 | 0.453 |
| ResNet 50 | 96.47 | 0.886 | 96.47 | 1.030 | 96.33 | 0.857 |
| Preactivation ResNet 164 | 96.20 | 2.539 | 96.20 | 1.079 | 96.08 | 0.921 |
| Wide ResNet 16x8 | 96.88 | 0.710 | 96.88 | 1.318 | 96.82 | 0.739 |
| Preactivation ResNet 18 | 96.15 | 1.574 | 96.15 | 0.645 | 96.05 | 1.096 |
| **average** | 96,587 | 1,336 | 96,587 | 1.05 | 96,494 | 0.847 |

| | **CIFAR100** | | | | | |
| | uncalibrated | | Temp Scal | | BNN | |
| | Acc | ECE | Acc | ECE | Acc | ECE |
| --- | --- | --- | --- | --- | --- | --- |
| WideResNet 28x10 | 80.39 | 4.853 | 80.39 | 4.276 | 77.59 | 2.456 |
| DenseNet 121 | 78.8 | 8.724 | 78.8 | 3.476 | 75.9 | 2.534 |
| ResNet 101 | 72 | 11.413 | 72 | 1.533 | 68.7 | 1.612 |
| VGG 19 | 72.7 | 17.631 | 72.7 | 4.798 | 71.94 | 6 |
| Preactivation ResNet 18 | 76.6 | 10.780 | 76.9 | 3.152 | 74.3 | 1.763 |
| Preactivation ResNet 164 | 73.28 | 15.754 | 73.28 | 2.046 | 70.77 | 1.461 |
| ResNext 29_8x16 | 77.88 | 9.678 | 77.88 | 2.811 | 73.97 | 2.581 |
| DenseNet 169 | 79.05 | 8.883 | 79.05 | 3.758 | 75.58 | 2.393 |
| Wide ResNet 40x10 | 76.74 | 14.767 | 76.74 | 3.765 | 76.17 | 1.876 |
| **average** | 76,36 | 11,387 | 76,36 | 3,291 | 73,88 | 2,520 |

| | **GENDER** | | | | | |
| | uncalibrated | | Temp Scal | | BNN | |
| | Acc | ECE | Acc | ECE | Acc | ECE |
| --- | --- | --- | --- | --- | --- | --- |
| VGG-19 | 90.60 | 8.08 | 90.60 | 3.96 | 90.50 | 2.70 |
| DenseNet 121 | 90.035 | 8.803 | 90.035 | 3.077 | 89.961 | 1.547 |
| ResNet 18 | 90.42 | 8.45 | 90.42 | 3.8 | 90.44 | 3.082 |
| **average** | 90.352 | 8.444 | 90.352 | 3.612 | 90.30 | 2.443 |

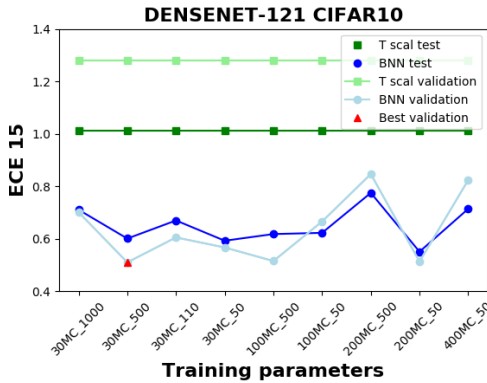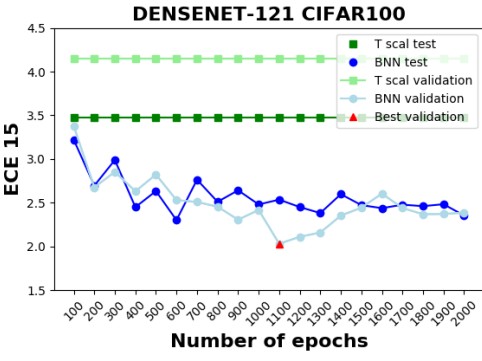

Figure 3: This figures compares the ECE performance for TS and BNN in test and validation. On the left (CIFAR10) we show the performance of different training parameters. For example 30MC_500 means that the ELBO was optimized using 30 MC steps to estimate the expectation and 500 epochs of Adam optimization. On the right (CIFAR100) we show the performance of a BNN trained with different number of epochs up to 2000, showing the robustness against the course of learning.

more complex logit space distribution (100 dimensional in CIFAR100) we could get better accuracy and better ECE increasing the expressiveness of the model. On the other hand in simpler logit space distribution (2 dimensional in GENDER) we found that the expressiveness must be reduced.

Finally, we realized that the BNNs are suitable and robust in the experiments carried out. In many experiments we found that all the tested configurations clearly outperform TS, as an example see figure 3.

## 6 DISCUSSION AND CONTRIBUTIONS

There is an increasing interest in adjusting confidence in deep learning, as this models are now becoming part of complex decision systems and critical applications. In the machine learning community there are two main approaches for reducing over-confidence, each one with its own pros and cons: implicit/online and explicit/offline. An implicit method aim at obtaining calibrated distributions directly at the output of the model, while explicit methods post-process the output of the model to be calibrated. Moreover both approaches can use either point estimate or Bayesian probabilistic models.

Explicit approaches have several advantages. First, one can calibrate pre-softmax values provided by other practitioners. Therefore, privacy concerns regarding the model or the original data used to train that model are considered. Second, explicit approaches can be combined with implicit ones (Lakshminarayanan et al., 2017; Seo et al., 2018; Kumar et al., 2018; Chen et al., 2018; DeVries & Taylor, 2018) to further improve calibration, as example see (Kumar et al., 2018; Lee et al., 2018), where TS is used within implicit approaches. Third, one can use impractical models applied directly to a deep model in this offline stage, e.g Bayesian Neural or Gaussian Processed. Fourth, we do not need deep architectures for the Bayesian stage as the input includes the already learned representation of the DNN. This stage only focus on adjusting probabilities. A two layer BNN is unable to reach the same accuracy as a deep convolutional model by its own, however, combined with it, can yield to state-of-the-art accuracy and calibration results, as we showed. Fifth, models to be calibrated does not need to be retrained. This easily let us calibrate, as example, models that make use of pretrained DNN (transfer learning applications). Sixth, any probabilistic model can be calibrated: CNN, LSTM-RNN, BLSTM-RNN, SVM, network ensembles... Seventh, some implicit methods, such as Gal & Ghahramani (2016); Seo et al. (2018) require us to train our deep models with Dropout or stochastic depth, respectively, while ours is totally independent on how the deep model is trained. On the other hand, implicit approaches are less sensible to overfitting. Guo et al. (2017) shows that more complex models yield worse calibration performance. However, we have demonstrated that correctly managing uncertainty allow us using complex models for post-processing, improving

state-of-the-art explicit approaches, and getting competitive results with the most recent published implicit ones.

Regarding the calibration performance applied to deep learning models. Our method reaches competitive results with state-of-the-art implicit approaches on deep learning models (Seo et al., 2018; Kumar et al., 2018), and outperforms other proposed explicit (Guo et al., 2017) and implicit (Tran et al., 2018) techniques. In fact, Kumar et al. (2018) obtain competitive results when combining their implicit method with TS, which again shows that offline calibration is a desirable and flexible choice to be combined with implicit models. In fact, we have been able to apply BNNs to a task of interest for the machine learning community, which is the main criticism to these techniques. Kuleshov et al. (2018), which propose a procedure for calibrating Bayesian algorithms only for regression problems, and Lakshminarayanan et al. (2017) argue that a Bayesian treatment do not output calibrated distributions, as the Bayesian deep learning has several restrictions that the machine learning community is trying to overcome. However, this work demonstrates that if we let the major complexity of the task to a deep model, a simple Bayesian approach can adjust probabilities in an efficient way. As shown in our github, 2-layers Bayesian neural nets are enough to adjust probabilities.

In terms of efficiency our BNN method presents several benefits in comparison to other Bayesian or point estimate methods. Although making predictions is more expensive than with TS, this predictions can be fully parallelized, as noted above, computing predictions in only one step. Moreover, a forward through a deep model and a shallow BNN is less computational expensive than a forward through a deep Bayesian convolutional model that requires several forward (and backward) for test (and training). For instance we have models based on BNNs (Gal & Ghahramani, 2015) and based on Gaussian processes (Tran et al., 2018; Milios et al., 2018). Network ensembles (Lakshminarayanan et al., 2017) reduce overconfidence and output calibrated distributions, but it is not measure in a deep model application, and only compared to Monte Carlo Dropout and to the number of ensembles. Ensembles can be also paralellized but in case of deep learning models, which is our field of study, the performance is compromised by the deepness of the different ensembles. As the authors state, computation restriction arises when evaluation of ensembles is done on ImageNet with the Inception network. Our model not only uses shallow neural nets but is only compromised by the number of classes of the task to be performed, and not by the complexity of the task at hand, as once we are able to reach a good accuracy we only focus on adjusting probabilities. Other implicit approaches such as Seo et al. (2018), that compute the cost to be optimized based on several predictions of the model, require to perform as many forwards per training samples as samples we want to estimate the cost parameter. This also compromise performance in deep models.

Finally, Lakshminarayanan et al. (2017) propose to train models with proper scoring rules, such as negative log-likelihood. However, as demonstrated by Guo et al. (2017) it is not clear if deep generative models trained with this criteria presents uncalibrated distributions, at least in implicit approaches.

## 7 CONCLUSION AND FUTURE WORK

This work has shown the many beneficial properties of offline calibration with a Bayesian reasoning. We open future perspectives which include: incorporate Bayesian improvements on the variational posterior with the objective of reducing topologies (efficiency in training and test time), better calibration and accuracy; be able to analyze how the logit dimension influences the expressiveness needed by the likelihood model and which key factors of Bayesian algorithms are critical for good performance; how can we model prior information on the parameters to yield better results; other offline approaches based on Gaussian processes, as example; incorporate training based on different proper scoring rules; measure robustness against adversarial examples; and implement these models in task where having good calibration is critical.

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

APPENDIX A: TOY EXAMPLE ON BAYESIAN CALIBRATION

In this case we consider a problem of density assignment. Formally, we have a set of data $\mathcal{O} = \{x_i\}_{i=1}^N, x \in \mathbb{R}^2$ belonging to class $c_1$, where its true distribution belongs to the family of bimodal Gaussian distributions. We want to assign a unimodal Gaussian parametric model $p(x|\theta)$ where $\theta = (\mu, \Sigma)$. Note that in this case the parametric model would be unable to recover the true distribution, which is likely to happen in deep learning models due to the complexity of the distributions these models cope with.

In the Bayesian framework the density is computed by averaging each possible model parameterized by $\theta$, using the posterior distribution computed from the observed data as the distribution over which we take the expectation:

$$p(x|\mathcal{O}, c_1) = \int d\theta_1 \, p(x|\theta_1) \cdot p(\theta_1|\mathcal{O}) \tag{5}$$

On the other hand, maximum likelihood would represent the data using the ML estimator $\theta^{ML}$. In this setting, it can be computed uniquely as the loss function has a global optimum.

Figure 4 shows an example of how model averaging improves the assignments of probabilities, and therefore model calibration, contrary to ML. The figure shows some data points generated by our training distribution, where the color of each point is different for each of the Gaussian mixtures. A point-estimate maximum likelihood (ML) model $p(x|\theta_1^{ML})$ (we use $\theta_1^{ML}$ to refer to the model assigned to $c_1$ datapoints) is fitted and represented as red contour lines. It can be seen that the ML model fails to accurately represent the true data distribution, although it can represent one of the two clusters of data moderately well. Two samples not observed in the training set are shown as a black and a gray dot. Also, different plausible likelihood models are represented in dashed contour plots.

We now assume that we have another set of data belonging to class $c_2$, but not represented in this figure as it is far in the data space. We fit $p(x|\theta_2^{ML})$ for this dataset. We assume the prior distribution over the classes to be equal for both classes, and based on Bayes theory decision, our decision rule is given by:

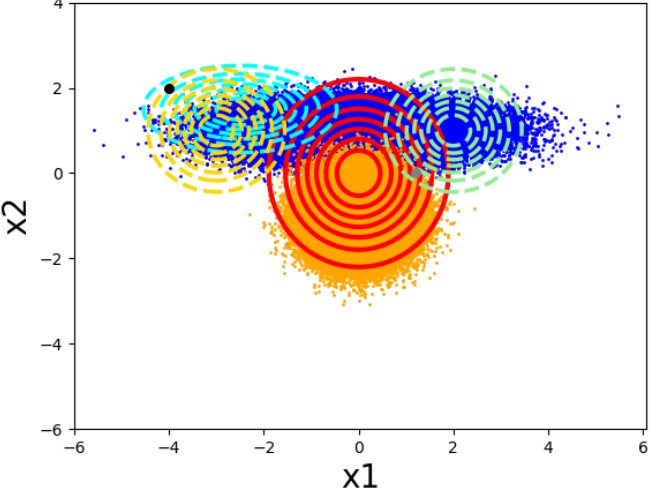

Figure 4: Bimodal distribution of 2-dimensional training data (orange and blue points) with Maximum Likelihood estimation of a Gaussian distribution (red contour) and other possible likelihood models explaining the data (dashed plots). Green and black dots represent data not seen in the training data, for which densities are to be assigned. Best viewed in color.

$$\frac{P(c_1|x)}{P(c_2|x)} = \frac{p(x|c_1)}{p(x|c_2)}, \tag{6}$$

where for generality we do not explicitly indicate if the model $p(x|c)$ is computed in the ML or Bayesian setting. As long as $p(x|c_1) > p(x|c_2)$ we will assign $c_1$ to a given sample $x$. It is clearly seen that for both the black and gray test samples, the density assigned by $p(x|c_1)$ is greater and thus these samples are assigned to class $c_1$. In fact, if both samples belong to this class we will have a perfect performance in terms of accuracy.

However, although the black dot is correctly assigned to $c_1$, the red ML model assigns extremely low density to it which is undesirable, as it has been actually generated like the rest of the data. This is not a desired behaviour, since it is in fact likely to belong to the blue component of the distribution as it is close to blue training samples in the data space. For that reason, if we compute probabilities under this model, the ultimate confidence would not reflect the true underlying process, and the calibration of the model will be affected. This effect is what we argue is happening in a classification framework: although we correctly choose the class (the cluster $c_1$) the ML model does not assign a correct probability.

On the other hand, if we take average of all the different models parameterized by $\theta = (\mu, \Sigma); \mu \in \mathbb{R}^2, \Sigma \in \mathbb{R}^{2\text{x}2}$ (see dashed plots in the figure), the density assigned to the black dot would be raised by some of the models that explain the blue data points. The importance given to each likelihood is given by the posterior $p(\theta|\mathcal{O})$. Therefore, other possibilities apart from the ML density will be considered, and thus we will be better modelling the probabilistic information. For an exact theoretical example on this same density estimation problem see (Minka, 2001). There you can find the exact posterior distribution using non-informative priors on the parameters.

