# OpenReview forum: "Offline Deep models calibration with bayesian neural networks"
_ICLR.cc/2019/Conference_

### Official Review · AnonReviewer1 · 2018-10-28
**Poorly Written Paper with Unconvincing Contribution**

**Rating:** 3
**Confidence:** 4

**Review:**

-- Paper summary --

The primary goal of this paper is to investigate the suitability of BNNs for carrying out post-calibration on trained deep learning models. The results are compared to equivalent models calibrated using temperature scaling, and the proposed technique is shown to yield superior uncertainty calibration.

-- General Commentary --

The overall goal of this work is rather modest and the scope of the evaluation is limited. While not without challenges, carrying out offline calibration as a corrective measure is a simpler problem to tackle than developing well-calibrated models upfront, and limiting the comparison to just one other post-calibration method greatly narrows the overall vision such a paper should have. For instance, isn’t post-calibration more likely to result in overfitting than a model that is implicitly calibrated at training time?

I have plenty of concerns with the submission itself, listed below:

- First and foremost, the paper is full of typos and grammatical errors. I genuinely struggled to read the paper end-to-end without being continually distracted by these issues. While some mistakes may indeed be genuine, others are only there due to sheer negligence and because the authors didn’t properly check the paper before submission.

- While the overall objective of this work (i.e. improving calibration of deep models) is clearly established, the overall presentation of ideas is very muddled and I initially struggled to properly understand what’s being proposed.  A simple diagram or illustration would have clarified some of the notation at the very least.

- The sloppiness in the presentation is also manifested in other ways. For example, in Figure 1, the plots should be individually titled (‘uncalibrated’, ‘temp-scal' and ‘BNN') in order to immediately distinguish between them; instead, all this information is contained in the caption whereas it could just as easily have been added to the plot.

- As alluded to earlier, I am disappointed by the lack of scope in the paper. The experimental evaluation should have been widened to include direct comparisons against BNN models which one might expect to be slightly better-calibrated upfront. There has also been significant interest in improving the calibration of deep models by stacking different architectures in such a way that the model is implicitly calibrated at training time. Examples of such papers include ‘Adversarial Examples, Uncertainty, and Transfer Testing Robustness in Gaussian Process Hybrid Deep Networks’ (Bradshaw et al, 2017), and ‘Calibrating Deep Convolutional Gaussian Processes’ (Tran et al, 2018). The deep kernel learning schemes developed by Wilson et al. also discuss similar hybrid models.

- With reference to the papers cited above, one possible extension the authors could consider is to use a Gaussian process for post-calibration instead of a BNN, although I suspect this may have already been investigated in the past. In any case, this warrants further discussion.

- I can’t disentangle the two contributions listed at the bottom of Pg 2 and the top of Pg 3. There is no theoretical evaluation of the ‘alternative hypothesis’ being mentioned, and the investigation is entirely limited to the offline setting, so I’m not entirely sure what distinction the authors are trying to make here.

- In the same section, the authors then remark that ‘Our results open new perspectives to improve the variational approximation…’ and ‘we believe our results might foster further research in…’, before proceeding to list a dozen or so papers which might be inspired by this work. However, I can’t really see how the single contribution being presented in this paper can have significant impact on the related work. I encourage the authors to substantiate their claims with more concrete examples rather than simply include vague mentions of other papers.

- The structure and content of Section 3 is quite perplexing. Effectively, up until Equation 3, the authors are simply restating how to use VI for BNNs, with no mention whatsoever of how this fits in the storyline of model calibration. Whereas such a section should have contained novel methodology and/or intuition, the only reference to using BNNs for post-calibration is found in a small paragraph at the end of Pg 4, before immediately proceeding to the Experiments section. Once again, this makes any contributions of the paper unclear and inconclusive. Spurious comments such as the inconsequential connection to MDL further accentuate the paper’s lack of identity and focus.

- There are also some problematic technical details in this section, such as the definitive choice of using a two-layered BNN with no justification whatsoever. It is well known than plain BNNs also struggle to deliver well-calibrated outputs, and yet the authors immediately settle on a two-layered fully-connected network without stopping to consider whether some other network configuration or initialisation scheme might be more appropriate. Some introspection is later given in the experiment accompanied by Figure 2, but the analysis carried out there is just not sufficient.

- There are some instances where the authors use text while in math mode, which gives poor formatting as exemplified by ‘conf’ in Equation 4.

- Referring to ‘datasets’ as ‘databases’ in Section 4.1 is unusual. Some of the commentary in this subsection is also very difficult to interpret. For example, what is meant by ‘uses BNNs’? Does this mean that a BNN appears in the model being calibrated or is this referring to the BNN used to carry out calibration? The majority of these ambiguous statements could have been avoided had more care been given to checking the paper properly before submission.

- In their discussion of the results, the authors state that ‘We cannot conclude that BNNs are calibrating at the cost of losing accuracy’, which I consider to be an overly sunny view of the results. Even if minor, a dip in accuracy is observed in almost every example provided in the Experiments section, dropping as much as 3% for CIFAR-100. Given that calibration is the primary focus of this paper, it might also be worth including another metic for validating this criteria, such as the Brier score.

-- Recommendation --

Unfortunately, the material presented here is neither significant enough nor sufficiently explored to spark much interest. The overall scope of the paper is disappointingly limited, while novel ideas and design choices are poorly motivated and communicated throughout. This submission feels rushed and incomplete, and consequently well below the conference’s standards.

Pros/Cons summary:

+  The proposal yields good results in the provided experiments
-   Minor contributions that are not convincing enough
-   Muddled presentation of ideas
-   Dubious or weakly motivated design choices
-   Poorly written with plenty of typos
-   Difficult to follow

---

> ### Author Response · Authors · 2018-11-12
> **Comments on first review**
>
> Thanks for your review. We first you encourage to read the comment we post in this review with the title GENERAL COMMENTS. There, we have answered important issues commonly highlighted by more than one reviewer.
>
> We thank the reviewer for all the comments. We will immediately take your recommendations on figure formats, correct typos, etc.
>
> We apologize for the presentation and writing of the paper. You are totally right that this might be successfully corrected in future versions of the article, in order to make the reading more understandable.
>
> We found most of the rest of your points interesting to be discussed, and that is why we have incorporated them in the general comments in our separate post. Thanks for pointing all this, we think these kind of discussions help to make our approach stronger.
>
> About your point on the accuracy degradation of our proposed model, you are right. However, we did not see strong evidence of that, mainly because accuracy degradation is not present significantly in all the results presented. Moreover, please consider that our calibration is performed on very state of the art results in an off-line manner while other approaches mentioned in the literature calibrate models that are far from that state of the art. We are currently exploring approaches to solve this issue, obtaining promising results. We are also extending our experimental results with more datasets and tasks.

---

> ### Comment · AnonReviewer1 · 2018-11-26
> **Post Rebuttal**
>
> I thank the authors for their detailed response to all reviewer comments and providing additional explanations about their method. Unfortunately, I tend to agree with other reviewers that the contribution isn’t significant enough in isolation, and requires a broader and more extensive experimental evaluation in order to make up for the lack of theoretical innovation.
> Consequently, my score remains unchanged.

---

> > ### Author Response · Authors · 2018-11-26
> > **New submission**
> >
> > We have significantly changed our submission. We hope that our contribution seems more clear in its new form. We have:
> >
> > -rewrite most of the paper
> > -give an insight on why Bayesian models are a good choice for calibration
> > -discuss our method more clearly towards other approaches

---

### Official Review · AnonReviewer2 · 2018-11-02
**Confusing and unsatisfactory**

**Rating:** 3
**Confidence:** 4

**Review:**

This paper proposes the use of Bayesian inference techniques to mitigate the issues of miscalibration of modern Deep and Conv Nets.

The presentation form of the paper is unsatisfactory. The paper seems to imply that Bayesian Deep Nets are used to calibrate Deep/Conv Nets, so I was expecting something like post-calibration using Bayesian Deep Nets. After reading through the paper a few times, it seems that the Authors are proposing the use of Bayesian inference techniques to infer parameters of Deep/Conv Nets in order to improve their calibration compared to non-Bayesian counterparts. This is the only contribution of the paper, and I believe it is insufficient. Guo et al., (2017) already points out that regularization of modern Deep/Conv nets improves calibration, so the fact that Bayesian Deep/Conv Nets are calibrated is not surprising, giving that the prior over the parameters act as a regularizer.

It is surprising to see ECE values above one - unless these have been scaled by a factor of 100 - but this is not mentioned anywhere.

Previous work shows that Monte Carlo Dropout for Conv Nets offers well calibrated predictions (https://arxiv.org/abs/1805.10522), so I think a comparison against this inference method should be included in the paper.

The paper makes a number of imprecise claims/statements. A few examples:

- "Bayesian statistics make use of the predictive distribution to infer a random variable by computing the expected value of all the possible likelihood distributions. This is done under the posterior distribution of the likelihood parameters" - very unclear and imprecise explanation of Bayesian inference

- "When using neural networks to model the likelihood" - the likelihood is a function of the labels given model parameters

---

> ### Author Response · Authors · 2018-11-12
> **Comment on first review**
>
> Thanks for your review. We first you encourage to read the comment we post in this review with the title GENERAL COMMENTS. There, we have answered important issues commonly highlighted by more than one reviewer.
>
> In order to make our work clearer, in the overall answer you can find a much wider explanation. There, you will find that we did not calibrated directly the deep model, but their outputs, and this is what make our method more efficient and flexible than other approaches. We apologize for not being so clear in the submitted paper.
>
> On the other hand, maximum posterior maximization also induces a prior over the parameters for regularization. In this case the calibration is achieved as we compute an expected value using the posterior (or its approximation) of the parameters. This posterior includes the prior and the likelihood. However, we did not take the maximum of it, but just a weighted average, and we believe that is this average what gives us confidence calibrated outputs, not the fact that the prior is acting as a regularizer. Additionally, we have provided in the general comments a discussion on why this approach can give us calibrated probabilities.
>
> Regarding ECE, yes, it is given as a percentage. Although we stated that in the caption of the tables, we will make it more clear in later versions of the paper.
>
> We will make clearer definitions for: “Bayesian Statistics make use of the predictive...”. Regarding the other comments by the reviewer, the likelihood, in this case, is a function of the labels, given model parameters and data. We use neural networks to parameterize this likelihood, but instead of training a point-estimated neural network, we train a distribution on model parameters. Maybe we should change the word “model” by the word “parametrize” in order to make the whole explanation clearer. Thanks for your fruitful suggestions in this sense.

---

> > ### Comment · AnonReviewer2 · 2018-11-22
> > **Thanks for the rebuttal**
> >
> > Many thanks for the response to our reviews. I still think that the paper adds little to the literature on the topic and that it requires a lot of work on the writing to be accepted in its current form.

---

> > > ### Author Response · Authors · 2018-11-26
> > > **New submission**
> > >
> > > We have significantly changed our submission. We hope that our contribution seems more clear in its new form. We have:
> > >
> > > -rewrite most of the paper
> > > -give an insight on why Bayesian models are a good choice for calibration
> > > -discuss our method more clearly towards other approaches

---

### Official Review · AnonReviewer3 · 2018-11-03
**A Bayesian method to improve probability calibration in deep neural nets, but lacks insights about why the method works well.**

**Rating:** 4
**Confidence:** 4

**Review:**

This paper presents an approach for calibrating the predictions of deep neural networks. The idea is quite simple and straightforward - simply use a more expressive model (Bayesian neural network with amortized inference). Surprisingly, the results show that this simple approach outperforms many of the recent approaches, such as those based on temperature scaling. A trick that they use is to control the KL term in the approximation of the ELBO using a hyperparameter (but this has been used in prior work on Bayesian neural nets).

Overall, the idea as such is not that novel (just applying a Bayesian neural network with amortized inference) but the results look quite impressive. However, although the paper seems to advocate that a simple Bayesian neural network is enough to get well-calibrated probabilistic predictions, recent work has shown that even Bayesian uncertainties may be inaccurate, especially in case of model mis-specifications or due to the use of approximate inference. For example, see  "Accurate Uncertainties for Deep Learning Using Calibrated Regression" ( Kuleshov et al, 2018).

It is quite surprising that the proposed approach works so well but there isn't much of an insight as to why it works well. Is it the amortized inference that helps, or something else? I think a more detailed analysis needs to be done. Even some empirical analysis that, for example, shows that using MCMC gives inferior results than amortized inference would help here.

Besides, I would also like to point out that there is some recent work on trainable calibration measures. See "Trainable Calibration Measures For Neural Networks From Kernel Mean Embeddings" (Kumar et al, 2018). It would be good to discuss this.

Overall, the paper has a rather straightforward idea which seems to give good results. However, it doesn't offer any new insights as to why it works, especially since recent work, such as the one I mentioned above, has shown that taking a simple Bayesian approach that provides uncertainty estimates doesn't quite address the problem being studied here.

---

> ### Author Response · Authors · 2018-11-12
> **Comment on first review**
>
> Thanks for your review. We first you encourage to read the comment we post in this review with the title GENERAL COMMENTS. There, we have answered important issues commonly highlighted by more than one reviewer.
>
> We have checked your suggested article, "Accurate Uncertainties for Deep Learning Using Calibrated Regression" (Kuleshov et al, 2018). We think that in this work the Bayesian approach they used is obtained by Monte Carlo Dropout. However, as argued by the original authors in the appendix section 5.1.1, this method yields outputs that are not calibrated. This can explain our calibration improvement using Bayesian techniques. We do not use Monte Carlo Dropout to get estimates of the predictive distribution. We use a separate Bayesian model instead.
>
> With respect to the comparison with MCMC algorithm, we respectfully disagree on the use of that technique, because MCMC algorithms are expensive when having big datasets. Moreover, MCMC algorithms have shown poorer performance than approximate Bayesian inference for this kind of task. However, as we discuss in section 5, there is recent work that shows that approximate Bayesian inference leads to an additional gap in the bound. The good point of our approach is that we can incorporate the solutions proposed in those works to our method. Any Bayesian improvement can be incorporated to our approach, and we believe that this is one of its strengths.
>
> We will incorporate this in the discussion thanks for that: "Trainable Calibration Measures for Neural Networks from Kernel Mean Embeddings" (Kumar et al, 2018). However, if you check the results of that paper, our approach is competitive with them, and even better in some cases.  Additionally, this work applies TS to their method to improve calibration, so again our method can be also applied instead of TS.

---

### Author Response · Authors · 2018-10-02
**Little mistake in the abstract**

We found a little mistake in the abstract. Where it says: "...., or to directly
train a calibrated deep convolutional model with Monte Carlo Dropout approximations, among others..."

we shoud say: " ...., or to directly train a calibrated deep convolutional model contrary to Monte Carlo Dropout approximations, among others..."

Instead of "with" place "contrary to" .

---

### Public Comment · ~Oliver_Jack_Goldstein1 · 2018-11-02
**Calibration of neural networks**

How relevant is your research on reducing the ECE, when clear issues exist with respect to the problem of adversarial attacks. By increasing the class ratio of adversarial examples in the test data, your calibration metrics become meaningless (in that they will score close to zero). How do you respond to the criticism of this line of work? Have you considered the role of adversarial examples?

---

> ### Author Response · Authors · 2018-11-12
> **We will consider this performance measure**
>
> Dear Oliver, thanks for your comment.
>
> Actually we have not measured our method against adversarial examples. However, we agree with you in that such evaluation can be powerful.  For that reason, we exposed the next hypothesis on what could happen with adversarial samples, so we can get feedback and suggestions to design a possible experiment. We should remark that we believe that this evaluation can be tricky, as although one can train the deep model using adversarial examples to make it robust against those, there exists lots of adversarial perturbations. However, let’s consider the worst case by assuming that our model is not trained nor validated with adversarial examples. From the point of view of temperature scaling (TS) we can argue the next things. We first suppose that, by training with adversarial examples, our accuracy dramatically changes from 90% to 20%.
>
> TS: The only thing TS can do is, by definition, modifying confidence as accuracy remains fixed. The perfect calibration would yield a 20% confidence for the selected sample. In this ideal case, if we have an adversarial example at test time with 0,2 confidence, it is our choice to decide whether we trust that selection or not. In the wort case, maybe our TS model outputs 0.8% and in that case we would have a really bad performance. If we get 0.0 confidence (that would be great) our model would still be uncalibrated and we would make wrong decisions anyway, both on normal and adversarial examples.
>
> BNN: This case is different. As we argue in the overall answer to the reviewers and will introduce in the paper, Bayesian models have the ability of modifying the confidence, depending on the sample. If a sample is not represented by our training data, a Bayesian model ideally outputs a uniform distribution meaning something like: “I do not know anything about this”. In that case our accuracy would be near to 1/C (where C is the number of classes) and the same for our confidence estimates. This would mean perfect calibration. In this case if the output of our model is 1/C we can decide not do trust our prediction. On the other hand, an adversarial example is computed from a point estimate network. There are lots of works that evaluate Bayesian models against adversarial examples showing that Bayesian models are robust against them. This is because Bayesian models may consider other possible likelihood models to which this sample might not be adversarial, and, although one of the model (the point estimate) get a really bad confidence, the average with the others might correct this, and thus even correct the accuracy. On the calibration part, which affects the confidence assigned, we could expect that the probability assigned (for the reasons exposed in the overall comment to reviewers) is also calibrated. So let suppose that the BNN corrects the 20% confidence to a 40%, that has to be done by correcting confidence (because we change our decision) so at least we would relax the confidence assigned by the point estimate network. Anyway, this are only hypothesis and we should check it experimentally. What we believe it is clear is that a Bayesian approach should suffer less from adversarial examples.

---

### Author Response · Authors · 2018-11-12
**GENERAL COMMENTS TO ALL THE REVIEWERS**



5.- Comparison with other methods


	We have compared our approach only with TS because some justified reasons, in our opinion. The first one is because TS is being adopted as a good calibration method. For example the works in https://arxiv.org/abs/1711.09325 or in http://proceedings.mlr.press/v80/kumar18a/kumar18a.pdf use TS to improve calibration. This also means that our method could be additionally combined with all the methods in those works, as well as others in the literature.

	On the other hand, we have seen that two reviewers have made emphasis on comparing ourselves with this work  https://arxiv.org/abs/1805.10522. Aiming at considering such a comparison, we must highlight that the proposed reference measures calibration on a limited amount of DNN models and tasks: particularly, it reports results on a resnet for the cifar10 task, and on a resnet for the cifar100 task. On the other hand, we have done 29 experiments in our present contribution. Moreover, our results are one order magnitude better in terms of ECE for both tasks, and accuracy is improved by a relative 15% in cifar10 and 3% in cifar100. In this conditions, we do not think that a comparison against this or other Gaussian-process-based approaches for calibration will be relevant for the community in the current state of the art.

	Additionally, we have searched in the literature for other works from the same authors, for example the following one to be published on NIPS 2018 https://arxiv.org/pdf/1805.10915.pdf . We have found both works really interesting and relevant. However, we have found some burdens or issues when facing the comparison with our work. For instance, the second mentioned work only address calibration comparing against another Gaussian based approach, and do not provide comparison to current state-of-the art, and moreover no code is provided. Additionally, the models presented in that work are not related to calibration of deep learning models which is our main contribution, so we consider rather irrelevant to find a comparison with this work.

	Anyway, if we check some of the works  that measure calibration like https://arxiv.org/pdf/1809.10877.pdf  (submitted to this ICLR, and thus impossible to be replicated for this submission) or http://proceedings.mlr.press/v80/kumar18a/kumar18a.pdf (proposed by one of the reviewers) one can easily see that our results are competitive with, or even better than them. This is in addition to the fact that these approaches are implicit, which supports our previous statements on that issue.  Moreover, the last one of these works( (http://proceedings.mlr.press/v80/kumar18a/kumar18a.pdf ) is only competitive with our approach when combining TS with their implicit calibration approach, which supports our observation that having good offline approaches combined with online approaches is desirable.

	Finally, we respectfully offer the reviewers the possibility to understand that comparison with other approaches in the state of the art should be aimed at the better-performing ones. There are lots of works that try to reduce overconfident probabilities (we cite them in our related work) but most of them either do not provide code or do not measure calibration. On the other hand, implementing and running implicit approaches to compare against can have a high cost in computation and time if no reproducible code is available. Mostly every implementation involves a high cost of implementation and experimental validation.


In summary, we believe that our contribution provides a new approach with outstanding performance for offline approaches, and competitive results with implicit approaches. In addition, our approach can be combined with implicit models. We believe that our response will contribute to better explain our point, and we apologize for the previous confusion, since we agree with the reviewers that we did not express ourselves in a fully clear way.

---

### Author Response · Authors · 2018-11-12
**GENERAL COMMENTS TO ALL THE REVIEWERS**

4.- Offline approaches vs implicit approaches.

Other criticisms expressed by the reviewers can be divided into the following two questions:

First: Is it not better to have an implicit model directly calibrated rather than using post-processing?

It is interesting and our answer is yes, having models being directly calibrated is good. However, we believe that having the possibility of calibrating an already trained model, or even combining an implicitly calibrated model with a state of the art calibration technique is even better. As an example, this work http://proceedings.mlr.press/v80/kumar18a/kumar18a.pdf (that was suggested by one of the reviewers) combines TS with their method, and this combination further improves over the implicit method. Moreover, offline calibration is desirable, as it allows to take any extremely complex pretrained model (such as ImageNet, which not all of the community is able to train from scratch due to computational limitations), do some transfer learning and then calibrate the output. Moreover, take into consideration privacy concerns.

Related to this point, we have to consider the fact that implicit sampling-based algorithms are typically much more expensive to train/test that ours. It is the case of Bayesian convolutional neural networks, any Gaussian-process-based approach as the reviewers suggest (https://arxiv.org/abs/1805.10522) or even parallel submissions to this ICLR (where variance estimates need several forwards through the deep model  https://arxiv.org/pdf/1809.10877.pdf ). Therefore, in terms of computational efficiency, our Bayesian offline approach present clear advantages.

Second: Is it not true that post-processing tends to overfit?

About this second question, the one related to overfitting, we agree with the reviewer: post-processing tends to overfit and, that is why probably vector and matrix scaling worsened results over TS in Guo et al., 2017). However, our reproducible results still support the hypothesis in our work, stating that “as long as uncertainty is correctly modelled, high expressive models can be used to improve the task of calibration”.

---

### Author Response · Authors · 2018-11-12
**GENERAL COMMENTS TO ALL THE REVIEWERS**

First of all, thanks for your comments and discussion. We agree with all of you that the writing of our paper can be improved and we are rewriting it to make it clearer. We have realized that most of your comments come from the fact that the ideas within the paper are not clear enough, or not sufficiently justified or explained. We apologize for that.

1.-Reproducibility of our results:

We have seen that some comments question the reliability of our results, as they seem really good in comparison to the current state of the art. We remember that our code is available to replicate each of the experiments in our github: https://github.com/2019submission/bnn.2019 . There, you can find exact instructions to replicate each number of each table in our work. You can even find a bash script that, by running it, will run a complete experiment in a specific task.

2.-A clearer description of our approach:

The reviewers state that the description of our work is confusing. We apologize for this. We will try to explain it better in the following lines. We state that our approach works in an offline setting. This means that we work on two stages. The first stage comprises the training of a deep convolutional model, which includes the typical issues with computational burden, data handling and memory. This stage is aimed at increasing the accuracy, and it is not a contribution of our article, just the starting point. The output of these first stage, i.e. the convolutional model, use to be badly calibrated, as demonstrated by Guo et al., 2017  which is the work we compared against to.

In order to calibrate this model we consider the pre-softmax values. To this end, we make a forward through the model of the train, test and validation sets. This logit or pre-softmax which is the same input to Temperature Scaling (TS from now own) algorithm is the input to our Bayesian Neural Network (BNN from now own). We train our BNN to map this uncalibrated logit to a calibrated logit space. Note that this pre-softmax values can be provided by other practitioners therefore privacy concerns regarding the model or the original data used to train that model are also considered and this is one important benefit of the off-line approaches, ours in particular.


3.-Motivation and theoretical justification of our approach:

Another concern of the reviewers is about the justification of our approach. We agree that we were not sufficiently descriptive on this, and we apologise. Our hypothesis is: “as long as you manage uncertainty correctly you can use high expressive models to improve calibration”. Since we have found references in the literature where the community uses and justify a Bayesian approach to calibrate distributions, we thought that this would be enough to justify our basic idea. In this sense, we can try to enlighten this discussion by the following argument:

It is known that Bayesian approaches manage better the model uncertainty than other point-estimate approaches (i.e. Maximum Likelihood, Maximum Posterior...). In this sense, the Bayesian paradigm models a distribution by weighting different contributions of different parametric models. Consider the case of a MAP network (e.g., a typical deep convolutional model). This model explains the data based on a point-estimate training, this is, it will represent what is more likely to appear. Nowadays, this gives outstanding accuracy in classification tasks, but it is easy to be over-confident, i.e., outputting extreme probabilities, even for data that has conditions not unseen in the training set. This is very harmful for the calibration, because for unseen data, where the network is not confident, it should yield moderate probabilities, otherwise the classification errors will be more catastrophic. In other words, in tasks where calibration matters, a classification error has unequal consequences if the probabilities are moderate or extreme. Thus, if the conditions of the testing data are very different from those on the training data, what is likely to happen, a point estimate will not represent the data in the way it should, possibly leading to over- or under-estimation of probabilities. On the other hand, in the Bayesian framework, a posterior distribution on the parameters could consider networks explaining that part of the unseen space. By averaging different contributions, the model in fact relaxes probabilities on unseen data having different conditions than in training.

---

### Meta-Review · Area_Chair1 · 2018-12-14

**Confidence:** 4
**Recommendation:** Reject

**Metareview:**

Reviewers are in a consensus and recommended to reject after engaging with the authors. Please take reviewers' comments into consideration to improve your submission should you decide to resubmit.